# Automatic Shift Control of an Electric Motor Direct Drive for an Electric Loader

**Shaole Cai, Qihuai Chen, Tianliang Lin \*, Mingkai Xu and Haoling Ren**

College of Mechanical Engineering and Automation, Huaqiao University, Xiamen 361021, China;
caisl_justdoit@163.com (S.C.); chen.qihuai@163.com (Q.C.); 20014080067@stu.hqu.edu.cn (M.X.);
happyrhlly@126.com (H.R.)
\* Correspondence: ltlkxl@163.com

**Abstract:** Traditional loaders with engines present the drawbacks of high energy consumption and poor emissions performance. The usage of an electric motor instead of an engine in an electric loader can effectively improve energy efficiency and emissions. The loader is mainly used in the earthwork construction of unstructured roads. Compared to the automobile, during the working process of the loader, the load fluctuates violently, and the vibration is serious. A large torque range during operation, a wide speed range during transfer, and frequently switching gears to ensure power are required by the loader. Therefore, the automatic shift control strategy for an automobile cannot be well applied to the loader directly. In this paper, a novel distributed electric motor-driven loader in which the walking drive system and the hydraulic system is decoupled is studied. The shift rule of the electric loader is also studied. A comprehensive automatic shift control strategy considering power and economy is proposed. Simulations are carried out to verify the feasibility of the proposed control strategy. The results show that under the "V" cycle operation condition of the loader, the shift rule meets the control requirements and the shift effect is obvious and reasonable. In terms of transfer conditions, the proposed control strategy yields ideal power performance and energy savings.

**Keywords:** energy saving; loader; electrification; automatic shift

## 1. Introduction

The electric loader is widely used. However, the traditional loader, driven by an engine, has the drawbacks of high energy consumption and poor emissions performance. The electric loader uses the electric motor to replace the engine, which itself is of great significance to the improvement of energy efficiency and the realization of zero emissions. At present, electrification technology has been widely applied to the automotive industry. However, there are significant differences between loaders and automobiles in terms of structure, transmission mode, and working conditions. Therefore, further research should be carried out on the electrification technology of loaders [1].

Loaders often work on unstructured roads. Working conditions are complex and the load changes violently. In order to meet the needs of the shovel loading operation and the transfer operation, a large torque output range and a wide speed range are required. Meanwhile, frequent gear switching is needed (up to 1000 times per hour) during shovel loading operation to ensure the power demand. Therefore, automatic gear shifting is important in reducing the workload of the operator, ensuring the handling of the loader, and improving the energy efficiency.

In-depth research has been carried out on the traveling shifting system and control strategy of construction machinery driven by engines at home and abroad. Most of them have realized semi-automatic and full-automatic control. Komatsu adopted the K-ATOMICS electronic control system and the ECMV valve to control the upshift rule of clutch oil pressure, so as to reduce the shift impact, improve the service life of transmission structural parts, and improve the comfort of driving operators. It has been widely used

in construction machinery such as wheel loaders [2]. The TECNORD company proposed flexible shift technology. After the shift command was issued, the controller determined an optimal combination oil pressure curve, so as to control the proportional reducing valve to realize the soft combination and separation of the clutch and improve the shift quality. OH et al. established an adaptive multi observer system [3–5], and Walker et al. designed a joint extended Kalman filter and a double extended Kalman filter to estimate and track the transmission torque of the clutch during gear shifting [6]. Hu et al. solved the internal and external interference in the shift process and improved the robustness of the system based on the adaptive controller which is based on back-stepping [7]. Wu et al. (based on Hamilton Jacobi inequality [8]) and Kim et al. (based on position tracking [9]) solved the internal and external interference in the shift process and improved the robustness of the system. LIUGONG cooperated with TRICM to develop a special transmission for construction machinery with dual variable integration technology. An electro-hydraulic control system, which was composed of a transmission control unit (TCU)—the solenoid valve—and a shift control valve, is studied. Finally, the combination and separation of the clutch or brake were controlled by the reduced valve, which improved the timing and pressure of clutch action in the shift process to a certain extent, realizing the shift with a certain quality. Wang et al. took the evaluation index of shift quality as the optimization objective and proposed a multi-objective optimization algorithm according to parameters such as the charging and discharging rate and combination speed difference of the clutch or the brake. The correctness and effectiveness of the parameter design and optimization method were verified through experiments [10]. Gao et al. proposed a reduced order clutch pressure observer [11]. Chen et al. established a dynamic model to analyze the shift process and proposed adaptive control [12]. Li et al. used the HP adaptive Legendre Gauss Radau positive mating point method to optimize the clutch or brake oil pressure trajectory in the shift process [13].

At present, research on the gear shifting technology of loaders mainly focuses on the traditional powertrain system and hybrid power train system in which the hydraulic torque converter is employed to realize the adaptive matching of the speed and torque between the power source and the traveling transmission shaft. The research content was mainly focused on the improvement of energy efficiency and control performance of the hydraulic torque converter. In an electric loader, the engine is replaced by an electric motor. If the hydraulic torque converter is still retained, the current automatic shift control can be transplanted to some extent. However, due to the low efficiency of the hydraulic torque converter, the overall efficiency optimization space of the system is limited. In this paper, a novel distributed electric motor-driven loader, in which the walking drive system and the hydraulic working system is decoupled, is studied. In this system, the hydraulic torque converter is canceled. The electric motor, together with a shift gear box, are applied to obtain the driving capabilities of wide speed range and wide torque range drive at the same time. However, due to the cancellation of the hydraulic torque converter, ensuring power and economy in the process of automatic shift requires further study.

## 2. System Configuration and Shift System

### 2.1. System Configuration

The engine in the traditional loader needs to drive the traveling and hydraulic working devices at the same time. The hydraulic torque converter is employed to realize the automatic adaptability between the engine and the traveling transmission shaft. The stepless speed regulation and the improvement of overload capacity are achieved. Problems such as the locked rotor working condition of the loader are therefore solved. However, due to the low efficiency of the hydraulic torque converter, the overall efficiency of the machine as a whole is unsatisfactory. Besides, the advantages of the high efficiency of a pure electric drive cannot be fully realized, resulting in a significant decline in the endurance of the machine.

Compared to an engine, the electric motor has a high frequency response, high overload capacity, and superior speed and torque regulation, which is ideal for the redesign of the transmission system of an electric loader. In order to more fully realize the advantages of a pure electric drive, a distributed electric motor drive scheme is studied. A dual electric motor is used to replace the engine to drive the hydraulic system and the traveling transmission system, respectively.

The scheme of a working power assembly of a pure electric drive loader is given in Figure 1. As can be seen, the loader system is divided into the power unit, the driving unit, and the actuator. The power unit is composed of the power battery, the battery management system (BMS), and the power distribution unit (PDU). The power battery is the energy source for the loader and stores the possible recovered energy. BMS is used for the battery power on/off management, fault detection, etc. PDU is used for the multiactuator split current matching of the DC bus. The driving unit can be divided into the upper hydraulic driving system as well as the lower traveling drive system. The upper hydraulic system and the lower traveling drive system are driven by two independent electric motors, respectively. The traveling drive system of the loader is driven by the electric motor through the reducer, which then directs the transmission shaft to drive the tire to move. At the same time, the electric motor controller (MCU) controls the electrohydraulic shift system through the transmission control unit (TCU) to separate and combine the wet clutch in the reducer.

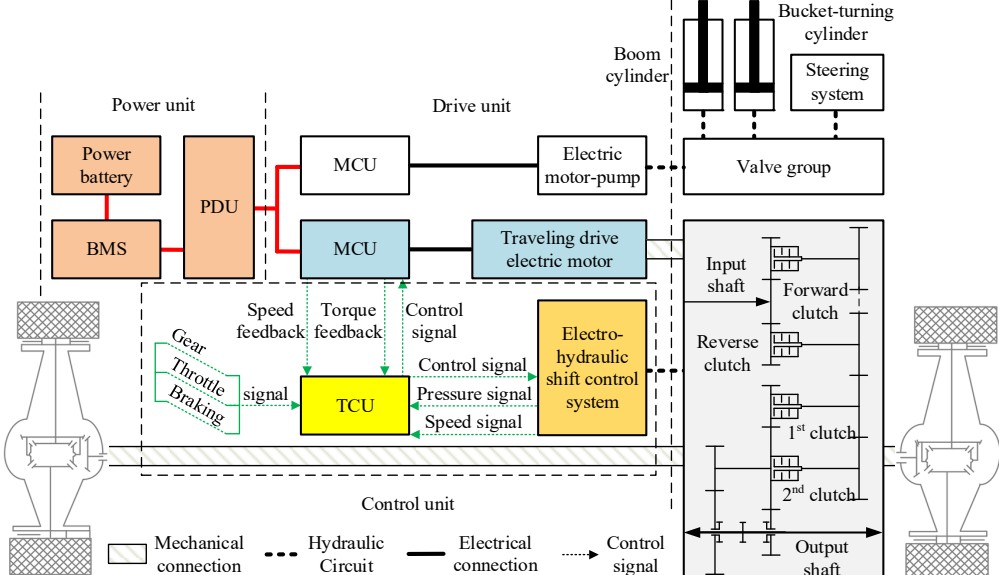

**Figure 1.** The scheme of a distributed electric motor drive system for a pure electric loader.

## 2.2. Shift System

To ensure the control performance, including the traveling speed during transfer operation and the driving torque during the shovel loading process of the electric loader, four degrees of freedom fixed axis power shift transmission are applied to the traveling system in the electric loader. The switching of high and low-speed modes by the combination of high and low speed meshing sleeves is then realized, followed by the direction control of the machine by the combination of forward and reverse gear direction gear clutches, and finally, the speed ratio control of the machine by the combination of first and second gear speed gear clutches is also realized.

The schematic diagram of an electro-hydraulic shift control system for the electric loader is given in Figure 2. The expansion and contraction of oil cylinder 22 is realized by controlling solenoid valves 21 and 23 to achieve the combination and separation of high-speed and low-speed meshing sleeves, so as to switch between high-speed mode and low-speed mode. By controlling solenoid valves 9 and 11, the combination and separation

of forward gear clutch oil cylinder 10 and reverse gear clutch oil cylinder 12 are controlled to realize the direction control of the machine. By controlling the proportional solenoid valves 13 and 17, the combined pressure of the first and second speed clutches 14 and 18 is controlled to realize the stable speed ratio control of the machine. In addition, throttle valves 16 and 20 are installed on the return oil circuit of the speed gear clutch to prevent excessive or insufficient overlap with the bonding clutch due to the quick pressure relief of the release clutch which will affect the shift quality [14].

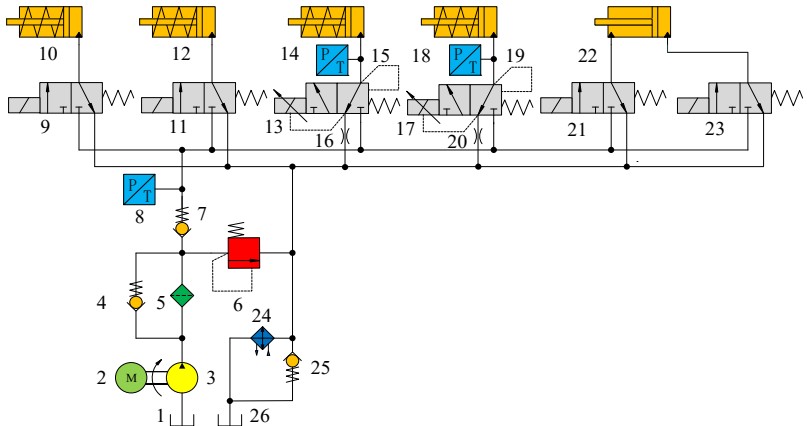

**Figure 2.** A schematic diagram of an electro-hydraulic shift control system of a pure electric loader. 1—Transmission box oil outlet; 2—Electric motor for the shift system; 3—Variable speed pump for the shift system; 4, 7, 25—Check valve; 5—Filter; 6—Overflow valve; 8. 15, 19—Pressure sensor; 9, 11, 21, 23—Solenoid valve; 10—Forward clutch; 12—Reverse clutch; 14—First clutch; 18—Second clutch; 13, 17—Proportional solenoid valve; 16, 20—Throttle valve; 22—High and low-speed mode switching cylinder; 24—Cooler; 26—Transmission lubrication return port.

## 3. Analysis of Automatic Shift Rule

According to different shift purposes and optimization objectives, the shift control can be divided into the optimal power shift and the optimal economic shift [15–17].

### 3.1. Optimal Power Shift Rule

The acceleration of the machine is selected as the objective function. Once the power transmission route of the whole machine is determined, the acceleration can be expressed as

$$a = f(v, \alpha), \tag{1}$$

where $a$ is the acceleration of whole machine, $f()$ is the whole machine acceleration formula, $v$ is the vehicle speed, and $\alpha$ is the accelerator pedal opening.

The optimal power shift rule is aimed at ensuring the optimal acceleration, which guarantees the optimal traction characteristics of the machine. However, the economy is unsatisfactory [18]. The optimal power shift rule can be deduced as the following: under a certain accelerator pedal opening, the intersection of the acceleration curves of two adjacent gears is taken as the shift point, which can be expressed as

$$a_\mathrm{n} = a_{n+1}, \tag{2}$$

where $a_\mathrm{n}$ and $a_{n+1}$ is the acceleration of the machine in gear $n$ and gear $n + 1$, respectively.

Taking the first gear and second gear switching under the low-speed mode of the transmission as an example, the design of the power shift rule can be described as follows:

(1) According to the external characteristic curve of the electric motor (models are given in Table 1), the speed torque curve of the drive motor under the opening of the 10%, 20%, ... , 100% accelerator pedal can be obtained, as shown in Figure 3.

(2) As seen in Figure 3, the driving force characteristic curve of the machine under different gears (Table 1) can be obtained. The intersection of two adjacent gears is taken as the shift point set. The shift point set under each accelerator pedal opening can form a rectangular optimal power shift area, as shown in Figure 4.

(3) According to the analysis above, in the optimal power shift area, two straight perpendicular lines are arbitrarily selected for the speed axis. The intersection of the straight line and the traction curve of the whole machine are connected. The best dynamic curve can then be obtained. In addition, when formulating the downshift rule, in order to avoid cyclic shift, the downshift curve is usually reduced by 2~8 km/h on the basis of the upshift curve. Due to the low overall speed of construction machinery, the downshift curve in low-speed mode is taken to reduce 2 km/h compared to the upshift curve, and then 3 km/h in high-speed mode. As a result, the optimal power rule curve can then be obtained, as seen in Figure 5.

**Table 1.** Parameters of the power battery, electric motor model, and mechanical transmission model.

| Component | | Parameter |
|---|---|---|
| Li-ion phosphate battery | | Nominal energy: 281.91 kWh<br>Nominal capacity: 456 Ah<br>Voltage: 480~700 V |
| Permanent magnet synchronous motor | | Rated power: 110 kW<br>Peak power: 220 kW<br>Rated speed: 1000 r/min<br>Rated torque: 1050 N·m<br>Rated voltage: 380 V |
| Shift box | Low-speed mode L | Forward F 1st gear: 3.488<br>Forward F 2nd gear: 1.806<br>Reverse R 1st gear: 3.488<br>Reverse R 2nd gear: 1.806 |
| | High-speed mode H | Forward F 1st gear: 1.126<br>Forward F 2nd gear: 0.583<br>Backward R 1st gear: 1.126<br>Reverse R 2nd gear: 0.583 |

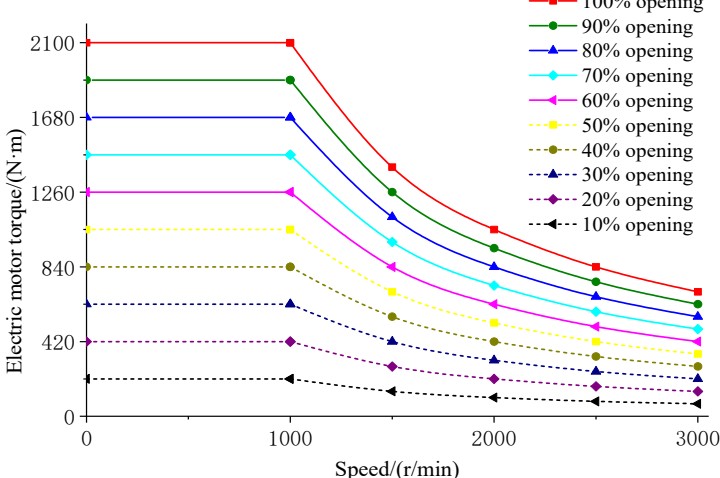

**Figure 3.** Speed torque curve of the 10%, 20%, . . . , 100% accelerator pedal opening of an electric motor.

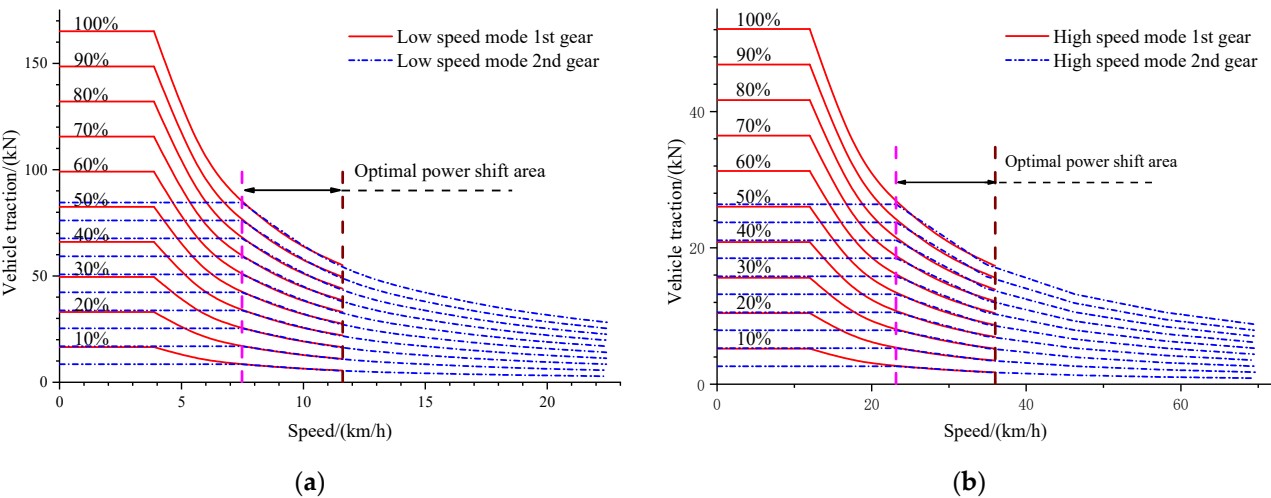

**Figure 4.** The driving force characteristic curve of the whole machine under different gears. (**a**) Low-speed mode; (**b**) High-speed mode.

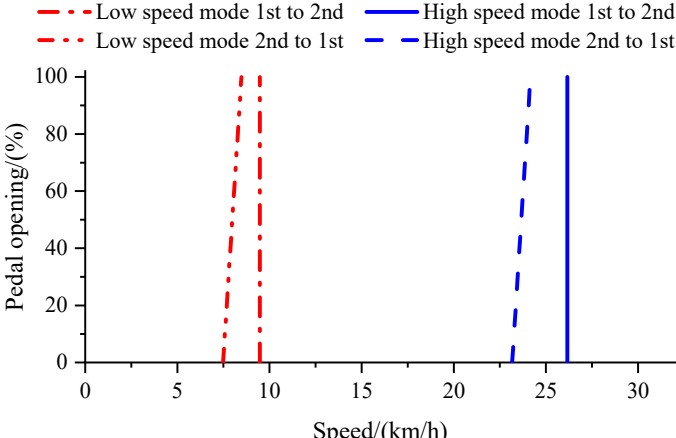

**Figure 5.** The optimal power shift rule curve.

### 3.2. Optimal Economic Shift Rule

The energy flow of the machine includes the conversion of electrochemical energy, the conversion of electrical energy to mechanical energy through the electric motor, and energy transmission of the mechanical transmission part. Thus, the energy efficiency of the electric loader traveling system can be expressed as

$$\eta_e = \eta_b \eta_m \eta_t \eta_z \eta_w \eta_r \eta_g \eta_d, \tag{3}$$

where $\eta_e$ is the total efficiency of the traveling system of the electric loader. $H_b$ is the efficiency of the battery and BMS. $H_m$ is the efficiency of the electric motor and MCU. $H_t$ is the efficiency of the transmission unit. $H_z$ is the efficiency of the main drive shaft. $H_w$ is the wheel side reduction transmission efficiency. $H_r$ is the road resistance utilization efficiency. $H_g$ is the gravity utilization efficiency. $H_d$ is the drive force efficiency.

When the power transmission system is determined, the shift rule mainly considers the steady-state situation of the loader. It is considered that $\eta_b$, $\eta_z$, $\eta_w$, $\eta_r$, $\eta_g$, and $\eta_d$ are unchanged or have little impact on the shift rule. The efficiency of transmission $\eta_t$ and the efficiency of the electric motor and its control unit $\eta_m$ under different gears are the main factors affecting the walking transmission efficiency of the electric loader. In this paper, $\eta_t$ and $\eta_m$ are adopted as the optimization goal of formulating the optimal economic shift rule.

(1) The efficiency map of the electric motor in the traveling system is given in Figure 6. Based on it, the speed efficiency curve of the electric motor under the opening of 10%, 20%, . . . , 100% of the accelerator pedal can be obtained (Figure 7).

(2) As shown in Figure 7, the machine speed efficiency characteristic curve under different gears can be obtained, the result of which is shown in Figure 8. If the efficiency curves of two adjacent gears intersect, the intersection point shall be taken as the shift point. If the efficiency curves of two adjacent gears do not intersect, the highest speed point of the efficiency curve of the lower gear shall be taken as the shift point.

(3) According to the above results, the shift points corresponding to the 10%, 20%, . . . , 100% accelerator pedal opening are connected in turn to form the optimal economic shift rule curve. Similarly, in order to avoid cyclic shift, when formulating the downshift curve, refer to the rule of the optimal power downshift rule. The downshift curve is usually reduced by 2~8 km/h on the basis of the upshift curve. The downshift and upshift curves with the optimal economic shift rule can be obtained by taking the shift curve as the center and translating 1~4 km/h to both sides, respectively. Due to the low overall speed of construction machinery, the downshift curve in low-speed mode is taken to reduce 2 km/h compared to the upshift curve, and 3 km/h in high-speed mode. As a result, the optimal economic shift rule curve is revealed, as seen in Figure 9.

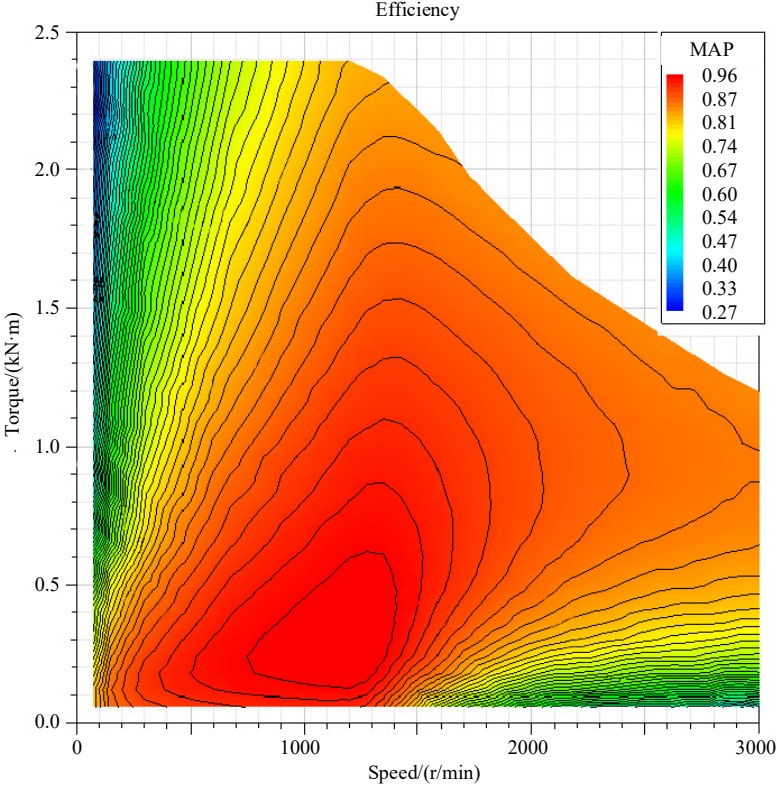

**Figure 6.** The efficiency map of an electric motor in the walking system.

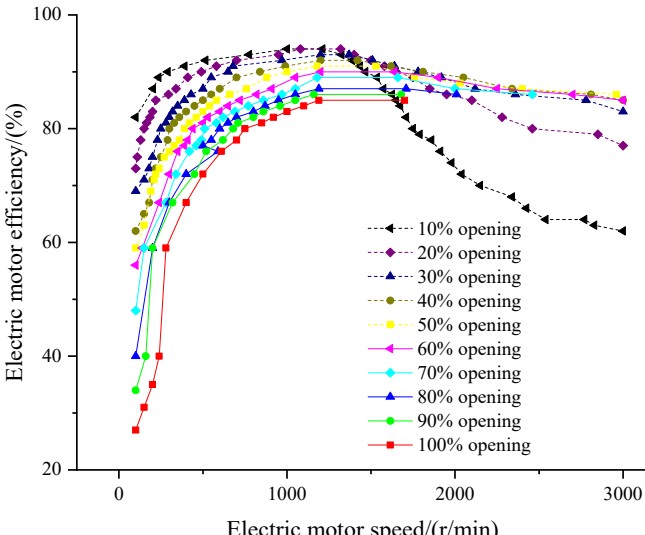

**Figure 7.** The speed efficiency curve of an electric motor under 10%, 20%, . . . , 100% of the accelerator pedal opening.

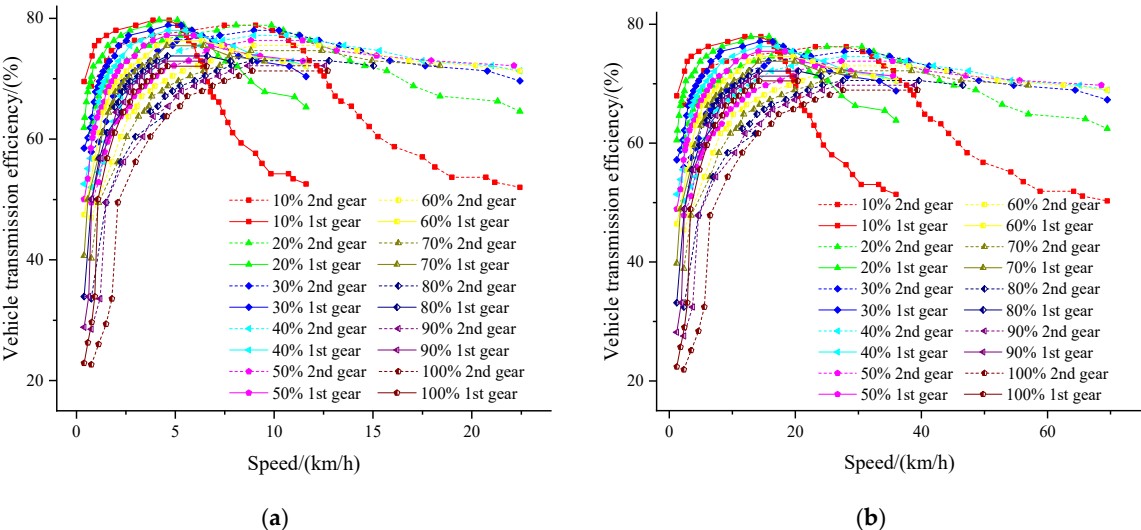

**Figure 8.** The machine speed efficiency characteristic curve under different gears. (**a**) Low-speed mode; (**b**) High-speed mode.

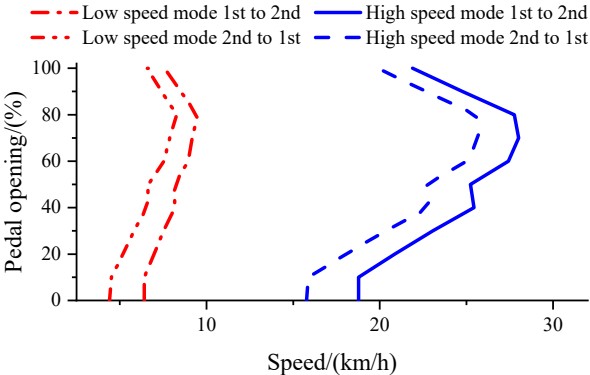

**Figure 9.** The optimal economy shift rule curve.

## 4. Comprehensive Shift Control Strategy

The application scenarios with different shift rules of the optimal power and the optimal economy are different. Under the working condition of the shovel loading, the loader operates in the power shift rule to ensure the power demand. Under the working condition of the transfer, the scene faced by the loader is also quite different from that of the automobile. The scene is more complex, which reduces the applicability of the simple economic shift rule. Therefore, a comprehensive shift rule for the optimal power and the optimal economy are both taken into account and studied.

Due to different purposes, there are differences between the preparation method and the rule itself in the shift rule of the optimal power and the optimal economy, as shown in Figure 10. It can be preliminarily concluded that the energy consumption difference of the optimal power and the optimal economic shift rule is obvious in the middle and low load area. The energy consumption difference in the middle and high load area is small. It meets the control demand of the driver which is "paying attention to economy under low and medium load, and power under medium and high load". In addition, it is also consistent with the changes of the complex road environment during the transfer of the loader. The flat, climbing, muddy, and rugged road types make the fluctuation amplitude and frequency of the accelerator pedal opening unstable compared to the automobile. The economic shift rule aiming at only low energy consumption is not so reasonable in the face of the large load. Although the power shift rule aiming at the pursuit of power can deal with the impact of complex and sudden environmental changes, the high energy consumption caused by it is unacceptable for operation.

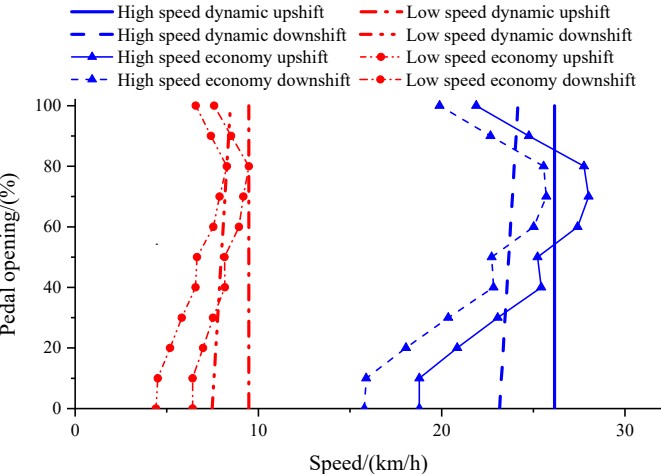

**Figure 10.** A comparison between optimal power and the economic shift rule.

Through the analysis of the differences between the optimal power and the optimal economic shift rule, in combination with the shift characteristics of the machine, a comprehensive shift rule considering both power and economy is proposed, which itself mainly considers economy under medium and low load and power under medium and high load. The control rule is given as follows:

(1) The power shift area is obtained according to the design method of the optimal power shift rule. The economic rule curve is obtained according to the design method of the optimal economic shift rule, as shown in Figure 10.

(2) According to the obtained power shift area and the economic rule curve, the overlapping between them can be obtained. If the economic shift curve overlaps with the power shift area, the highest load point of the overlapping part is taken as the dividing point between power and economy. The optimal economic shift rule is adopted in the area below the load value corresponding to the dividing point, and the optimal power shift rule is adopted in the area above the load value corresponding to the dividing

point. As shown in Figure 11, the analysis diagram of the comprehensive shift rule of high-speed mode is the same as that of the low-speed mode, which will not be repeated here.

(3) If there is no overlap between the optimal economic shift curve and the optimal power shift area, the point where the economic shift curve is nearest to the power shift area is taken as the dividing point. The optimal economic shift rule is adopted in the area below the load value corresponding to this point, and the optimal power shift rule is adopted in the area above the load value corresponding to the dividing point. The developed comprehensive shift rule curve is shown in Figure 12.

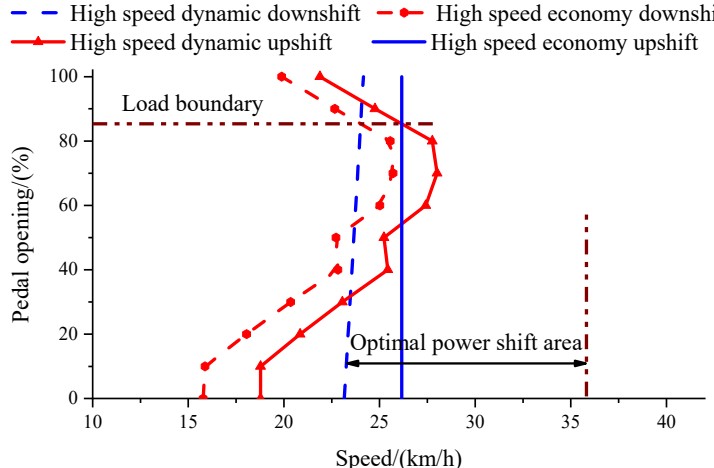

**Figure 11.** The comprehensive shift rule.

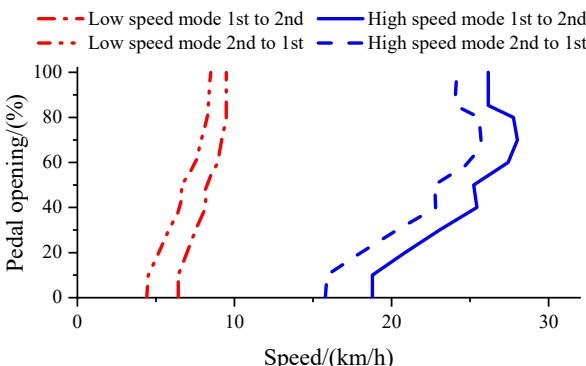

**Figure 12.** The comprehensive shift rule curve considering power and economy.

## 5. Simulation Research on Automatic Transmission System of Loader

In order to verify the rationality of the shift rule of the machine, co-simulations through Matlab/Simulink (2018b, MathWorks company, Natick, MA, USA) and AMESim (2020, Siemens Digital Industries Software, Plano, TX, USA) based on the actual operation of the loader are carried out.

### 5.1. Machine Model

According to the power transmission system and machine configuration of the electric loader, to simplify and facilitate the analysis, the influencing factors which are not related to the research objectives are eliminated. The modular models in AMESim are selected as far as possible. A 5-t electric loader is employed for simulation. Driver model, VCU model, power battery and electric motor model, transmission model, whole machine model, and joint simulation interface are selected. Among them, the cycle condition setting in the driver model is selected to be imported into the personal established condition. Energy

recovery during braking is not to be considered. Among them, the parameters of the power battery, electric motor model, and mechanical transmission model are shown in Table 1. The AMESim model of the automatic transmission shift rule is shown in Figure 13.

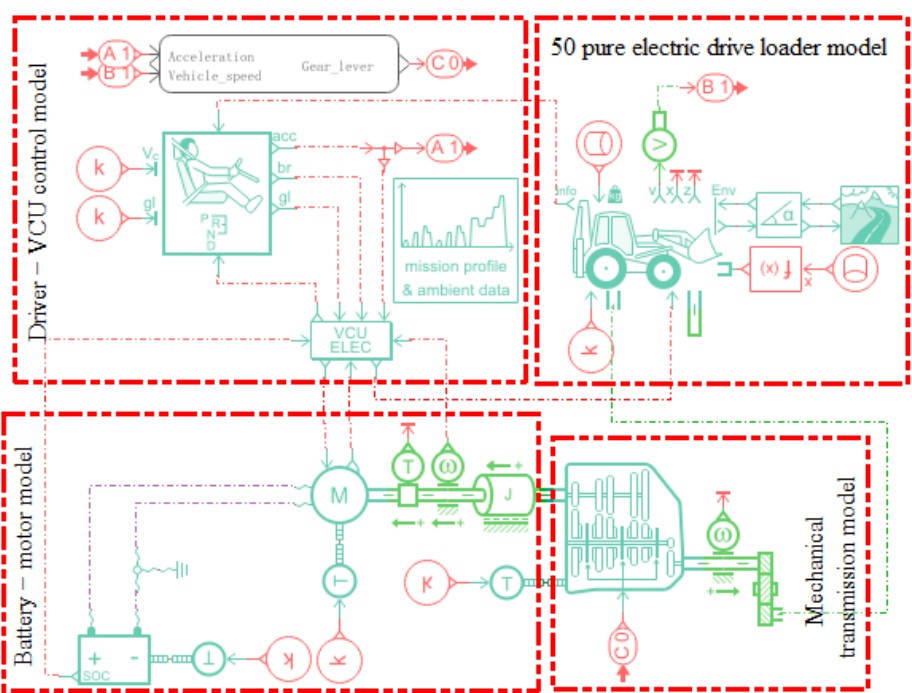

**Figure 13.** The automatic transmission shift rule based on the AMESim machine model.

*5.2. Automatic Shift Rule Control Model*

The automatic shift rule control model in Matlab/Simulink is shown in Figure 14. It mainly includes the shift rule model, the shift logic judgment model, and the joint simulation interface. Through the feedback of a loader running a state signal in AMESim, shift judgment is carried out according to the proposed shift rule in Simulink, so as to output the shift control command to the machine to complete automatic shift.

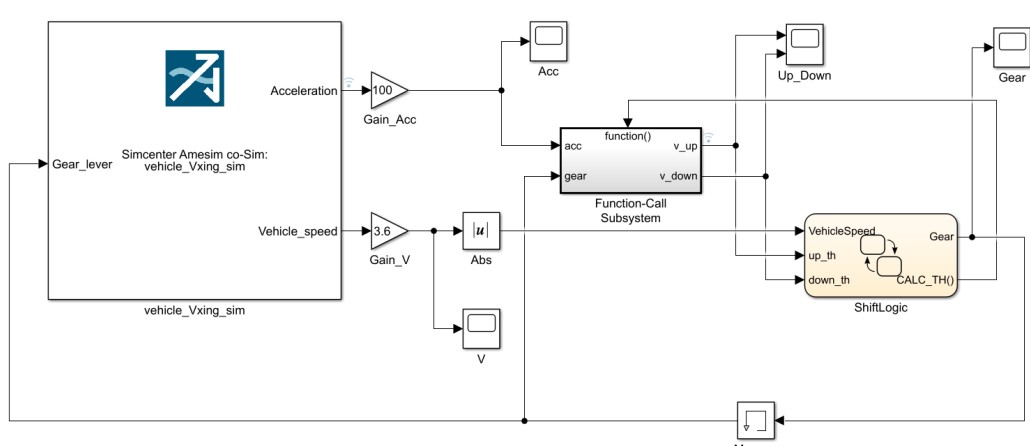

**Figure 14. A** Matlab/Simulink model of an automatic transmission shift rule control.

5.2.1. Shift Rule Model

As the core of the shift control strategy, the main purpose of the shift rule is to obtain the upshift speed and the downshift speed of the loader under different gears and different accelerator pedal openings, so as to provide the shift signal for the shift logic judgment model. The shift rule model includes multiple two-dimensional data tables. The

corresponding upshift speed and downshift speed are obtained by using the input vehicle speed and accelerator pedal opening. The shift rule model is shown in Figure 15.

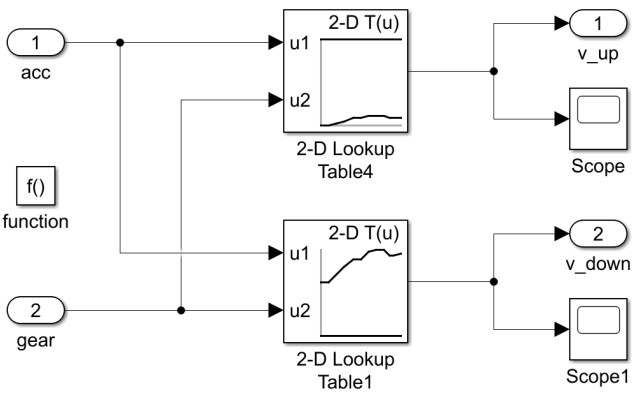

**Figure 15.** The shift rule model.

### 5.2.2. Shift Logic Model

Based on the event process, the shift logic sub model compares the current speed with the speed of the upshift and downshift in a stable state to determine whether a shift is required; it is a control logic module for multi-state mutual conversion. The shift logic judgment model is shown in Figure 16.

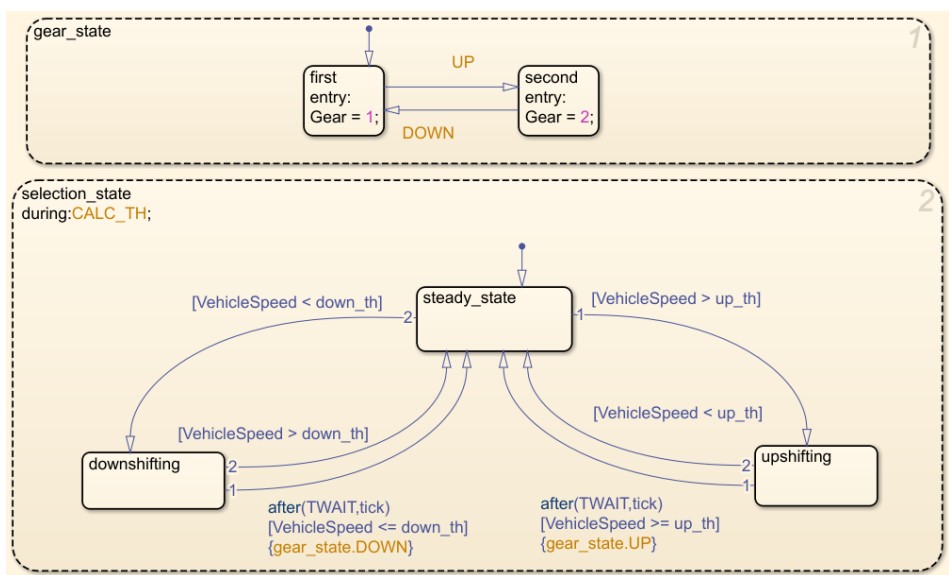

**Figure 16.** The shift logic model.

Gear_state and selection_state are a gear decision-making model and shift control model, respectively. Vehicle speed is the real-time vehicle speed. Up_th and down_th are the vehicle speed at the upshift point and downshift point, respectively. Up and down are the upshift and downshift state transition events, respectively. Upshifting, downshifting, and steady_state refer to upshift state, downshift state, and stable state, respectively. Gear is the output of the loader shift logic model.

If the vehicle speeds no longer meets the shift conditions, in the confirmation state, the model will ignore the shift and then switch back. This prevents unnecessary shifts due to noise conditions. If the shift condition is still valid for the duration of the scale, the model will transition through the lower intersection and propagate one of the shift events according to the current gear. The model is then reactivated after a transition through a central intersection. The shift event propagated to the status will activate the transition to the corresponding new gear. The subsystem activation function calc_th

controls the activation state of the shift law module. When the shift control module selection_state is activated, the function calc_th activates the shift law module to calculate the shift point speed.

### 5.3. Shift Rule Simulation Analysis

The simulation analysis of the loader shift rule consists of two parts: the shovel loading condition simulation and the transfer condition simulation. These mainly verify the accuracy and effectiveness of the established typical cycle conditions and the proposed automatic shift rule.

### 5.3.1. Shovel Loading Condition

The loader mainly adopts the optimal power shift rule to improve the dynamic performance of the machine, so as to ensure high operation efficiency. Since there is still no standard loader cycle, the correctness of the typical "V" cycle of the loader based on the test data is verified. Secondly, the accuracy of the built loader automatic transmission simulation model is verified. The shift rule results are analyzed.

### Simulation Results of Loader "V" Cycle

In the simulation model, the measured data of the typical "V" cycle working condition of the loader are imported. The driving speed data shown in Figure 17 are used as the target speed of the simulated machine. The resistance time data in Figure 18 is inserted to simulate the external load of the machine during shovel loading. The machine mass versus time data in Figure 19 is used to simulate the load of the machine during operation. The simulation results are shown in Figures 20 and 21.

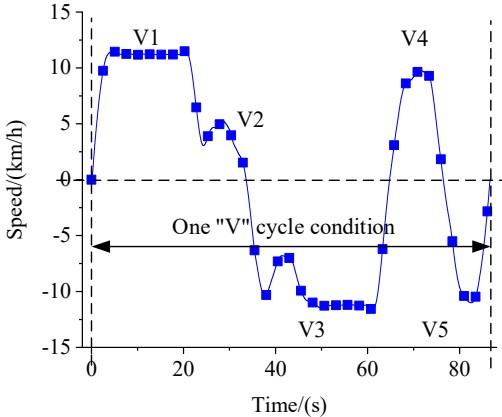

**Figure 17.** Data of the driving speed.

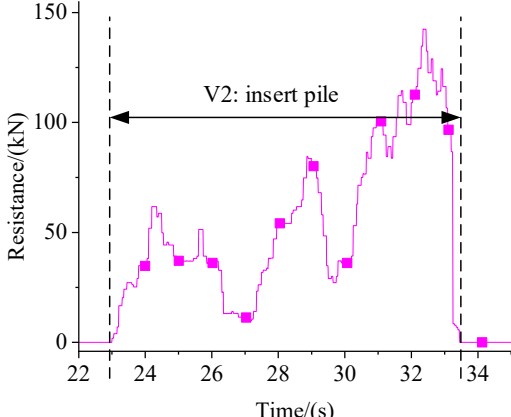

**Figure 18.** Data of the insertion resistance.

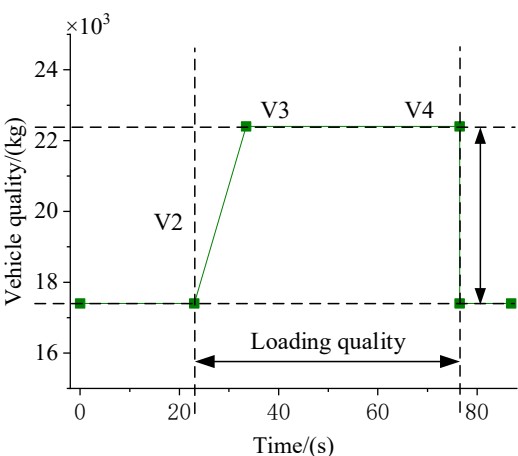

**Figure 19.** Machine mass–time data.

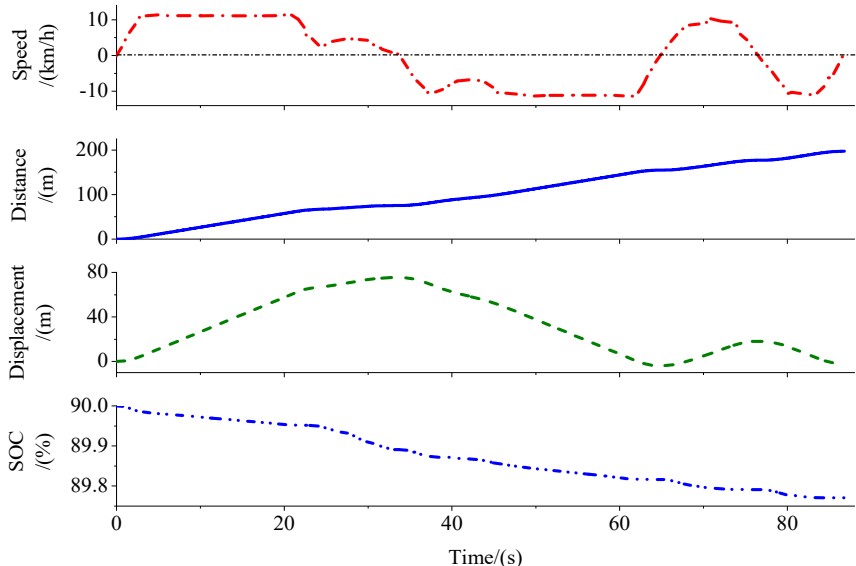

**Figure 20.** The simulation results of machine speed, driving distance, displacement, and SOC.

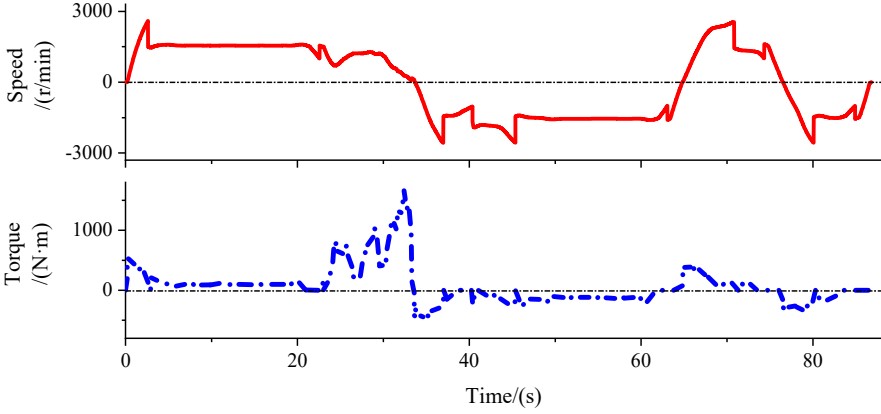

**Figure 21.** The simulation results of the electric motor.

At the same time, the load spectrum of the actual loader under the typical "V" cycle is measured, as shown in Figure 22. From the simulation results of the loader speed, driving distance, and displacement, it can be determined that the loader operates according to the established shift rule under a typical "V" cycle. The torque simulation results of

the traveling drive motor are basically consistent with the trend characteristics of each segment of the actual vehicle traction test time history given in Figure 22. When the loader enters the shovel loading stage, the load value reaches its maximum and the amplitude is consistent. In addition, the SOC of the power battery consumes 0.22% under a "V" cycle, which can be completed about 40 times in an hour, so the walking drive consumes about 24.8 kWh per hour.

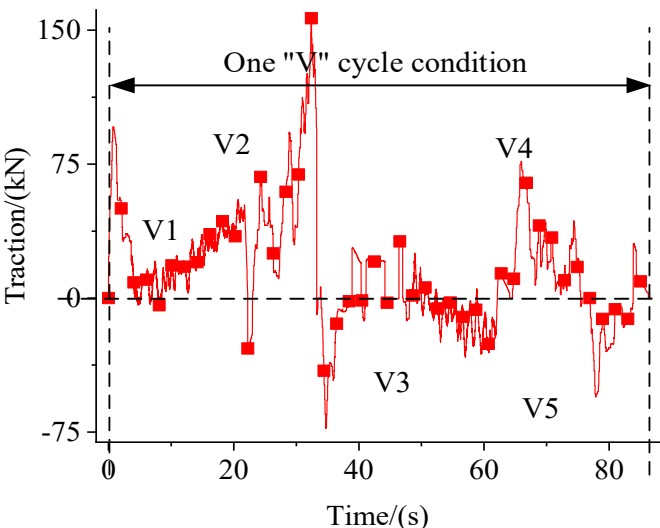

**Figure 22.** The load spectrum of the actual loader under the typical "V" cycle.

Simulation Results of Loader Shift Rule

The established "V" cycle condition is taken as the target condition of the machine simulation, and the proposed optimal power shift rule is simulated. The simulation results are shown in Figure 23.

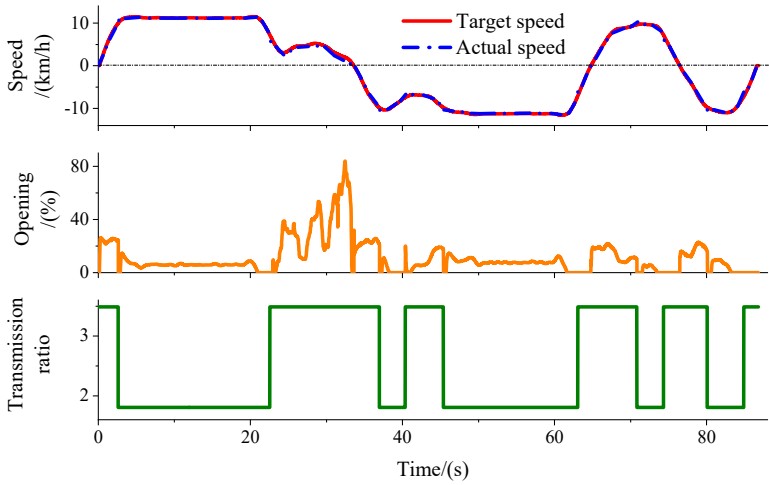

**Figure 23.** The simulation results of the shift rule under the "V" cycle condition.

It can be seen that the automatic transmission simulation model of the loader established in this paper can basically follow the simulation of the "V" cycle. It can be seen from the shift curve that in order to ensure the dynamic performance of the loader during operation, the shift effect is obvious and reasonable, which verifies the feasibility of the optimal power shift control strategy formulated in this paper for the working conditions of the loader.

From the simulation results of the speed curve, it can be seen that the simulation speed follows the target speed well, which shows that the automatic shift simulation model of the

loader established in this paper can simulate the "V" cycle condition, and the simulation model is basically accurate. It can be seen from the transmission ratio curve that the loader drives quickly to the material pile under the "V" cycle condition. In this process, when the speed reaches the upshift speed, the upshift is completed to ensure a higher speed. When arriving at the material pile, the vehicle speed drops rapidly due to the existence of resistance. The power output under the second gear is not enough to ensure the power required for shovel loading. The power is ensured by downshifting. Specifically, at 36.6 s, the loader enters the full load backward transportation stage. In this stage, the first gear starts, and the vehicle speed increases rapidly with sufficient power. After reaching the upshift speed, it rises to the second gear. Near 40 s, the vehicle load increases, and the vehicle speed decreases, realizing downshift and ensuring sufficient power. From 36.6 s to 64.8 s, the loader is in long-distance transportation, and it can be seen that it drives in second gear most of the time. After 64.8 s, the loader enters the full load forward transportation stage and into the no-load backward transportation stage at 76.5 s. These two stages also start in the first gear, change to the second gear when the speed rises, and then reduce to the first gear when the speed decreases. To summarize, the shift effect of the whole cycle is obvious and reasonable, which verifies the feasibility of the best dynamic shift control strategy formulated in this paper for the working conditions of the loader.

### 5.3.2. Transfer Conditions
### Economy Characteristic

As mentioned above, the energy consumption difference between the optimal power shift rule and the optimal economic shift rule is obvious in the middle and low load areas, while the difference is small in the middle and high load areas. In this paper, the economic performance simulation model of an electric loader is established. Since there is no simulation cycle condition suitable for the transfer condition of the loader, a CHTC-D condition of a Chinese commercial machine after proportion transformation which is consistent with the characteristics of the transfer condition of a loader is taken for simulation. Under this circumstance, the optimal power shift rule, the optimal economy shift rule, and the comprehensive shift rule are simulated and analyzed, respectively. The simulation results are shown in Figures 24–26, respectively.

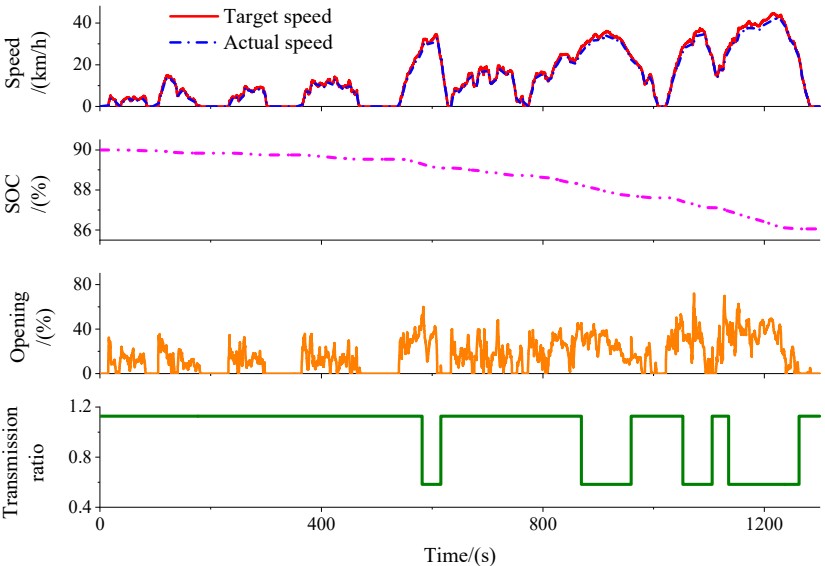

**Figure 24.** The simulation results of the energy consumption of the optimal power shift rule.

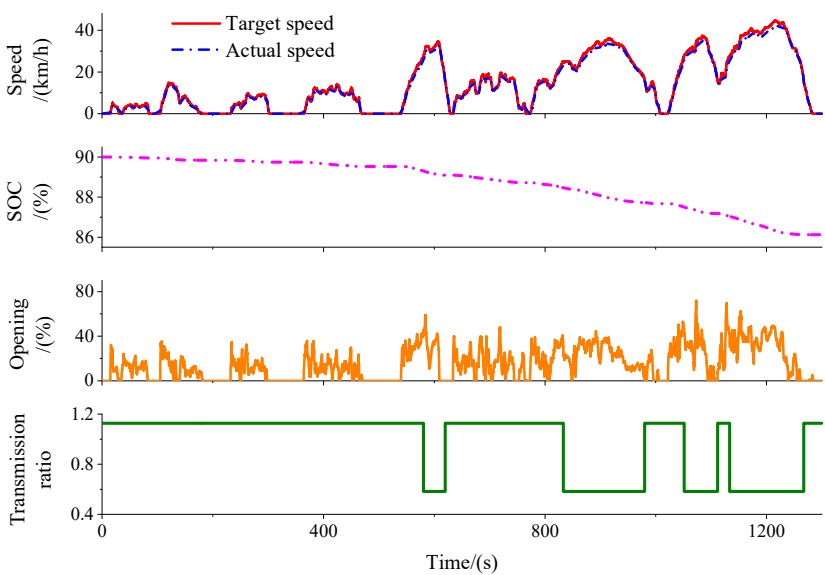

**Figure 25.** The simulation results of the energy consumption of the optimal economic shift rule.

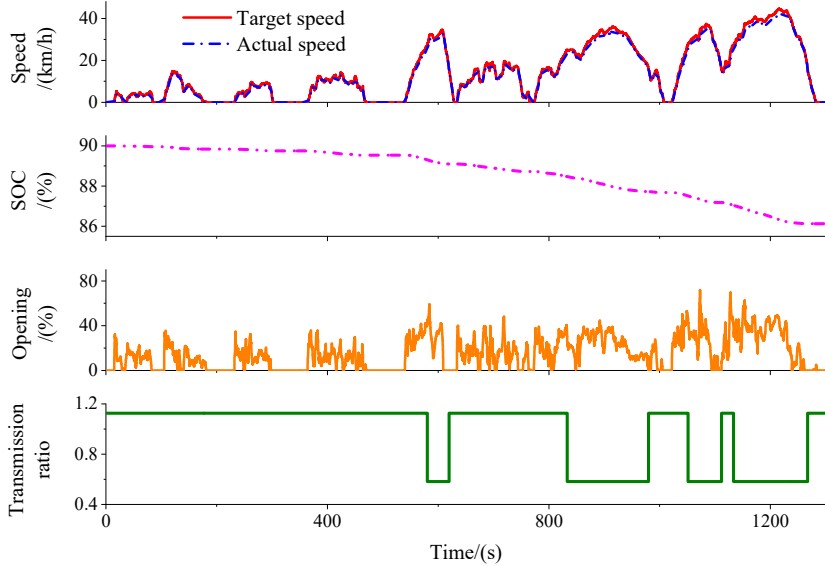

**Figure 26.** The simulation results of the energy consumption of the comprehensive shift rule.

It can be seen in Figure 27 that the whole cycle lasts for 1300 s. In terms of the energy consumption evaluation index, the simulation results of the comprehensive shift rule are consistent with the optimal economic shift rule. All of them shift actively. Besides, the energy consumption of the machine is reduced. The SOC of the optimal power shift rule decreases from 90.00% to 86.06%, and that of the comprehensive shift rule decreases from 90.00% to 86.13%, which is 0.07% less than that of the optimal power shift rule, about 0.20 kWh, accounting for 1.78% of the SOC consumed by the optimal power shift rule. The SOC difference is mainly composed of five parts marked in Figure 27, accounting for 4.34%, 61.20%, 23.95%, 4.20%, and 6.30% of the total reduction, respectively. Combined with the opening pedal curve, it can be obtained that the first, fourth, and fifth parts are medium and high load areas, and that their SOC consumption accounts for only 14.85% of the total consumption, while the second and third parts are medium and low load areas, and their SOC consumption accounts for 85.15% of the total consumption.

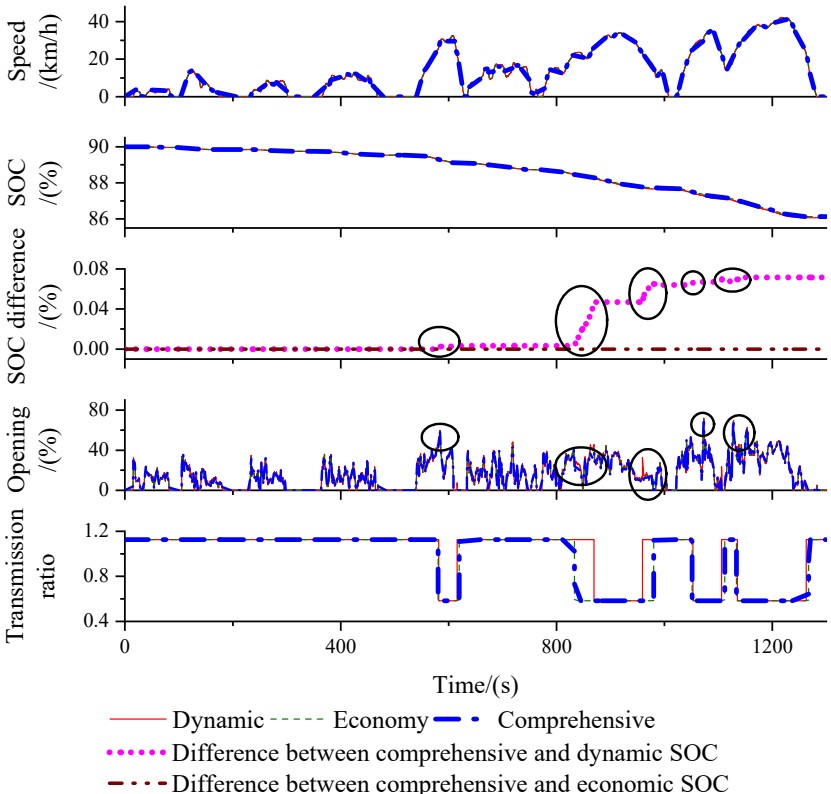

**Figure 27.** The simulation results of the energy consumption of three shift patterns.

Dynamic Performance

In this paper, the power performance simulation model of an electric loader is established. The acceleration performance is taken as the evaluation index of power performance. The optimal power shift rule, the optimal economy shifts rule, and the comprehensive shift rule are simulated and analyzed, respectively. The simulation results are shown in Figure 28.

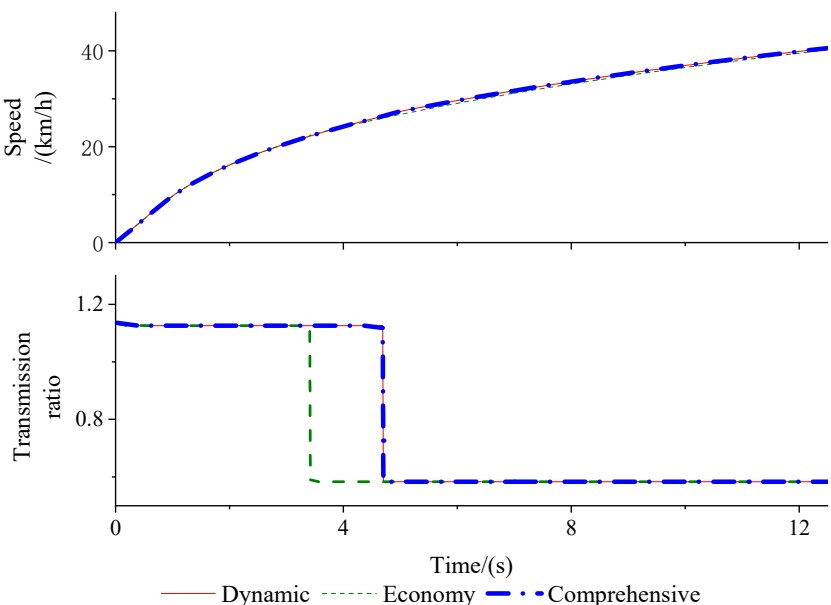

**Figure 28.** The simulation results of the acceleration performance of three shift rules.

As can be seen in Figure 28, in terms of acceleration performance evaluation index, the simulation results of the comprehensive shift rule are consistent with those of the optimal power shift rule. The shift occurs at 4.70 s, while the shift of the optimal economic shift rule occurs at 3.43 s. Since the loader does not have a complete set of acceleration performance evaluation indexes, the machine speed near the shift position where the comprehensive shift rule is completed is taken as the evaluation index. The acceleration time of the optimal power shift rule and the comprehensive shift rule from 0 to 27 km/h is 4.85 s, while the acceleration time of the best economic shift rule is 5.11 s. Contrastingly, the acceleration time of the optimal power shift rule and the comprehensive shift rule is 0.26 s less than that of the optimal economic shift rule from 0 to 27 km/h, accounting for 5.09% of the acceleration time of the optimal economic shift rule from 0 to 27 km/h. The acceleration performance gap gradually decreases after 27 km/h.

Therefore, simulation results of the economic and dynamic differences are in line with the above-mentioned optimal power shift rule. The economic gap between the optimal power shift rule and the optimal economic shift rule is obvious in the middle and low load areas and small in the middle and high load areas. The difference of power performance between 0~27 km/h is obvious, and then the gap gradually decreases. On the basis of the requirement of "low and medium load pays attention to economy and medium and high load pays attention to power" of the driver and operator of the loader under the condition of transfer transportation, the comprehensive shift rule not only has the power of the optimal power shift rule, but also considers the economy of the optimal economic shift rule. Finally, the results of the difference between economy and power are shown in Table 2.

**Table 2.** The results of the economic and dynamic differences.

| Parameter | | Dynamic Shift | Economical Shift | Comprehensive Shift |
|---|---|---|---|---|
| Economy difference | Energy consumption value/(%) | 3.94 | 3.87 | 3.87 |
| | Comprehensive difference value/(%) | 0.07 | 0 | - |
| | Proportion of comprehensive difference/(%) | 1.78 | 0 | - |
| Dynamic difference | Acceleration time/(s) | 4.85 | 5.11 | 4.85 |
| | Comprehensive difference value/(s) | 0 | 0.26 | - |
| | Proportion of comprehensive difference/(%) | 0 | 5.09 | - |

## 6. Conclusions

Traditional loaders have the drawbacks of high energy consumption and poor emissions performance. The use of an electric motor instead of an engine can effectively improve energy efficiency and emissions. However, there are significant differences in structure, working conditions, and operation requirements between loaders and automobiles. Therefore, the traditional automatic shift control strategy cannot be well transplanted to the loader. Aiming for a novel type of distributed motor-driven loader with decoupled walking drive and hydraulic working system, research on the automatic shift technology of an electric loader is carried out in this paper. The influence of the automatic shift rule on the power and the economy of an electric loader was analyzed. The comprehensive shift

control combining the optimal power shift rule and the optimal economy shift rule of an electric loader, according to the proposed walking power assembly scheme, was designed. Verification research was carried out to validate the effectiveness of the proposed system and the automatic shift rule.

**Author Contributions:** Conceptualization, Q.C. and T.L.; methodology, Q.C.; software, S.C.; validation, S.C., M.X. and H.R.; formal analysis, Q.C.; investigation, Q.C. and S.C.; resources, Q.C. and T.L.; data curation, S.C.; writing—original draft preparation, S.C.; writing—review and editing, Q.C. and T.L; visualization, H.R.; supervision, T.L.; project administration, T.L.; funding acquisition, Q.C. and T.L. All authors have read and agreed to the published version of the manuscript.

**Funding:** The research is funded by the National Key Research and Development program (2020YFB2009900), the National Natural Science Foundation of China (51905180), the Fujian University Industry University Research Joint Innovation Project Plan (2022H6007), Fuxiaquan National Independent Innovation Demonstration Zone System Innovation Platform (3502ZCQXT202002), Natural Science Foundation of Fujian Province (2021J01295)). This work has also been supported by Hitachi Construction Machinery Co., Ltd.

**Institutional Review Board Statement:** Not applicable.

**Informed Consent Statement:** Not applicable.

**Conflicts of Interest:** The authors declare no conflict of interest.

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
