# Peer review of "Automatic Shift Control of an Electric Motor Direct Drive for an Electric Loader"

_machines, doi:10.3390/machines10050403_

Round 1

Reviewer 1 Report

  1. Please carefully check the reference section. Authors name are not listed in the reference.
  2. Please revise the English. Currently, the text is very difficult to read. 
  3. Figure 6, values are difficult to read. 
  4. No comparative analysis is presented. Please compare the proposed logic based shifting rule with the existing literature. 
  5. Please improve the discussion part by providing table summarizing the results.  

Author Response

Many thanks for your comments and suggestions. All the comments and suggestion are incorporated into the revised manuscript. The modification of the manuscript is highlighted through yellow in the revised version. 

The detailed modification can be found in the attachment.

Reviewer 2 Report

The main question is: is there a good shift strategy, that has ideal power performance and energy saving, for an electric loader?

A combined optimization of gear ratios and shift control would be interesting, as in the presented analysis gear ratios were fixed. Moreover, it would be interesting to see how the differences, between the shift strategies, in terms of dynamic performance and energy consumption vary when different gearing is adopted. The author could consider, under the V-cycle, a shift strategy that reduces the number of shifts, to preserve the transmission.

Authors are asked to better comment Figure 1. Consider that the readers are not only expert in transmission. Split the figure in some parts if necessary for a better description.

Some other Figures might require a further comment.

Showing fewer curves in fig. 8 would be clearer. In fig. 26 the line "difference between comprehensive and dynamic SOC" is not clearly visible

I suggest a table summarising the results before the conclusions 

References: remove Author 1, Author 2....

Author Response

(The authors gave the same response as above.)

Reviewer 3 Report

There are several concerns and issues that need to addressed before making a final recommendation:

  1. Please include a clear problem statement. Try to be concise and include effort in developing your theoretical framework. 
  2. Lack of detailed analysis of the distributed electric motor drive system of pure electric loader as well as the detailed derivation of the proposed shift rule. For example, what is the mathematical representation of the system? No proof of stability is given in terms of the proposed shift rule.
  3. The major contribution of this paper is unclear. How is your proposed strategy differentiated from the existing ones? Can you please include some comparisons?
  4. I highly suggest that the authors reorganize the contents or improve the structure of this paper. Avoid lengthy I really feel it difficult to read this paper...

Author Response

(The authors gave the same response as above.)

Reviewer 4 Report

Eliminate typographical errors

In line 22 remove the numbers

In line 185 better define f

In line 285 the text below Figure 10 move

In line 311 the text below Figure 12 move

In line 340 Title hierarchy (5.2.1)

In line 349 Title hierarchy (5.2.2)

In line 383 Title hierarchy (5.3.1.1)

In line 406 Title hierarchy (5.3.1.2)

In line 418 Title hierarchy (5.3.2.1)

In line 451 Title hierarchy (5.3.2.2)

In Reference everywhere to remove the text "Author 1,2 ..."

Author Response

(The authors gave the same response as above.)

Round 2

Reviewer 1 Report

No more comments.

Author Response

Many thanks for your comments for the improvement of the manuscript.

Reviewer 3 Report

The authors stated "In terms of electrification, the research on automatic shift of electric loader, especially distributed electric loader, is still blank at home and abroad." in the cover letter which I disagree with. I can easily find a relevant research cited as follows "Li, Xuefei, et al. Operating Performance of Pure Electric Loaders with Different Types of Motors Based on Simulation Analysis. Energies 14.3 (2021): 617." Please comment on that and improve the quality of your manuscript in your final submission.

Author Response

Many thanks for your comments to improve the manuscript.
